# Physical Restraint Events in Psychiatric Hospitals in Hong Kong: A Cohort Register Study

**DOI:** 10.3390/ijerph19106032

**Published:** 2022-05-16

**Authors:** Maritta Välimäki, Yuen Ting Joyce Lam, Kirsi Hipp, Po Yee Ivy Cheng, Tony Ng, Glendy Ip, Paul Lee, Teris Cheung, Daniel Bressington, Tella Lantta

**Affiliations:** 1Xiangya Center for Evidence-Based Practice & Healthcare Innovation, Central South University, Changsha 410083, China; 2Department of Nursing Science, University of Turku, 20014 Turku, Finland; yutlam@utu.fi (Y.T.J.L.); kirsi.hipp@hamk.fi (K.H.); tejela@utu.fi (T.L.); 3School of Nursing, Hong Kong Polytechnic University, Kowloon, Hong Kong SAR, China; teris.cheung@polyu.edu.hk; 4Community Psychiatric Services, Pamela Youde Nethersole Eastern Hospital, Chai Wan, Hong Kong SAR, China; chengpyi@ha.org.hk (P.Y.I.C.); ncp977@ha.org.hk (T.N.); 5Central Nursing Division, Kwai Chung Hospital, Kwai Chung, Hong Kong SAR, China; glendyip@yahoo.com; 6Department of Health Sciences, University of Leicester, Leicester LE1 7RH, UK; paul.h.lee@leicester.ac.uk; 7College of Nursing and Midwifery, Charles Darwin University, Casuarina, Larrakia Country, Darwin NT 0810, Australia; daniel.bressington@cdu.edu.au

**Keywords:** physical restraint, coercion, psychiatric, Hong Kong, register, hospital

## Abstract

The need to better monitor coercion practices in psychiatric hospitals has been recognised. We aim to describe how physical restraint events occur in psychiatric hospitals and identify factors associated with physical-restraint use. A cohort register study was used. We analyzed physical restraint documents among 14 wards in two psychiatric hospitals in Hong Kong (1 July and 31 Dec. 2018). In total, 1798 incidents occurred (the rate of physical restraint event 0.43). Typically, physically restrained patients were in early middle-age, of both genders, diagnosed with schizophrenia-spectrum and other psychotic disorders, and admitted voluntarily. Alternate methods for physical restraint were reported, such as an explanation of the situation to the patients, time-out or sedation. A longer period of being physically restrained was associated with being male, aged ≥40 years, having involuntary status, and neurodevelopmental-disorder diagnosis. Our findings support a call for greater action to promote the best practices in managing patient aggression and decreasing the use of physical restraint in psychiatric wards. The reasons for the use of physical restraint, especially for those patients who are admitted to a psychiatric hospital on a voluntary basis and are diagnosed with neurodevelopmental disorders, needs to be better understood and analysed.

## 1. Introduction

Coercive interventions, such as physical restraint, are commonly used methods to manage patient aggressive behaviour in psychiatric treatment settings [1]. This is despite the trend of reducing the use of coercive methods in psychiatric hospitals and that physical restraint may constitute a violation of international human rights [2,3,4]. Concerns have particularly been raised about the use of mechanical restraints in psychiatric hospitals due to the potential severe physical [5] and psychological harm to patients [6] and a lack of evidence of its clinical benefit [7]. On the other hand, the use of physical restraint varies significantly between countries. Studies conducted in Wales, Ireland, Germany, and the Netherlands showed that the use of physical restraints on admitted patients ranged from 4.5 to 9.4% [8] while in four Pacific Rim countries (Australia, New Zealand, Japan, USA), the use of coercive interventions varied widely between 0.03 (New Zealand) to 98.9 (Japan) per million population per day [9]. In general, in psychiatric hospital wards in the UK, the incidence of the use of a mechanical restraint, which is a type of physical restraint, has been reported to be 0% [10]. As coercive methods and any form of physical restraint should be used as the last resort to protect the patient, staff or other patients from physical harm [11], more effort should be directed to understanding the events of physical restraint in psychiatric hospitals.

Physical restraint has been defined as any action or procedure that prevents a person’s free body movement to a position of choice and/or normal access to his/her body by the use of any method that is attached or adjacent to a person’s body and that he/she cannot control or remove easily [12]. Several guidelines have already been developed to guide physical-restraint practices in different international settings [13,14,15]. Most guidelines in psychiatric settings highlight the importance of respecting patient autonomy [15], patient safety [13,14,15], and staff training [13,14,15]. There has been a call to action for the systematic and open monitoring of the use of coercive methods in psychiatric hospitals globally [10]. An open policy of monitoring and recording coercive methods has already been implanted in many Western countries [16,17]. During recent years, a number of studies based on the monitoring system have reported that the use of coercive methods in Asian countries has increased. For example, in studies conducted in China, the use of mechanical restraints varied from 27.2% (*n* = 1364 patients) [18] to 51.3% in 160 patients [19]. Another study in Hong Kong showed that 39.7% out of 335 patients from four wards were restrained within the first week of admission [20]. In Taiwan, 29.5% (59 out of 200) of patients visiting psychiatric emergency services were restrained during their treatment period [21]. Ye et al. [22] concluded that the frequency of restraint was higher in China compared to the global average. It has also been reported that over half (61.2%, *n* = 129) of the psychiatric inpatients surveyed in Hong Kong reported traumatic experiences due to witnessing another patient being physically restrained and 41.1% reported being placed in restraints of any kind during their admission [23].

Typically, physical restrictions result from patient aggressive behaviour [17]. Weltern et al. [24] identified patient risk factors for aggressive behaviour such as the diagnosis of psychotic disorder or bipolar disorder, substance abuse, a history of aggression, and younger age, whereas ward risk factors were a higher bed occupancy, busy places on the ward, walking rounds, an unsafe environment, a restrictive environment, lack of structure in the day, smoking and lack of privacy [24]. Factors associated with a patient being subjected to physical restraint in Asia seem to be somewhat similar to those reported in Western studies [25]. A Chinese study with 160 psychiatric inpatients revealed that male gender, with less outpatient treatment prior to admission, a more frequent use of mood stabilizers, more aggressive behaviour prior to admission, and younger age were associated with an increased likelihood of being physically restraint during patients’ hospital stay [19]. A Hong Kong study also found that older patients admitted to a psychiatric hospital under ‘involuntary status’ and without psychiatric medication, with a history of violent behaviour but lacking a psychiatric diagnosis, were more likely to be restrained [20]. In China, after the implementation of the new mental health legislation, some new factors associated with the use of restraints have been investigated, such as unemployment, lower income, aggression in the past month, being admitted before the new legislation and poorer insight [18].

To date, many studies have been published on the use of coercive methods in high-income Western countries. However, less information is available related to patient coercion practices in low and middle-income countries, including Asia [26]. Existing research has focused mostly on the prevalence of seclusion and restraint, and factors associated with their use [19,20]. The circumstances of physical restraints, including when, why and its outcomes are also less studied topics. As the use of coercion practices is known to be affected by cultural, social, legal, and other contextual factors [27], research about physical restraint used in the Asian contexts is still needed. Therefore, in this study, we aimed to describe all patient physical restraints as they were documented in two psychiatric hospital medical records. Based on the existing literature, we assumed that patient-related factors, such as age, gender, legal status, and diagnosis could be associated with physical restraint incidents. Although the reasons for using physical restrictions in psychiatric hospitals may differ, the requirement remains that any forced intervention must be necessary, reasonable, and proportionate. The topic of physical restraint has been an interest of local, national, and international studies in recent decades. As far as we are aware, this is the first study in which register data has been used, even on a small scale, to understand the events of physical restraint, such as factors associated with patients being physically restrained in Hong Kong psychiatric hospitals. This study will therefore offer a good foundation for further analysis and practice-development projects on how patient care could be further developed in Asian psychiatric hospitals. The results could be used to identify possible gaps in treatment provisions [25] and to understand whether more targeted intervention might be needed and by whom to prevent the use of coercive measures in health services. In addition, the results might be used for staff education to identify any trends related to patient physical restraint practices. The study is part of a larger project aiming to understand patient aggressive behaviour and the use of coercive methods in Hong Kong psychiatric hospitals.

The overall aim of this study was to describe physical restraint incidents in two psychiatric hospitals as reported by hospital staff in the patient-restriction reporting system during hospital stay. The following research questions were specifically addressed:What are the general characteristics of the physical-restraint incidents?When did the events of physical restraint occur?What types of physical-restraint events occurred?What are the reasons for physical restraint and what factors are associated with these events?

## 2. Materials and Methods

### 2.1. Design and Setting

A cohort register study design was used to analyse treatment documents in two psychiatric hospitals. We included all recorded incidents of physical restraint as they occurred in the study wards between 1 July and 31 December 2018 in the analyses. In this study, physical restraint refers to the use of a physical or mechanical device to limit or prevent movement of the whole or a portion of the patient’s body as a means of controlling his or her physical activities [28]. The study was conducted in two psychiatric hospitals under the Hospital Authority (HA) in Hong Kong. All seven hospital clusters were invited and two clusters were willing to join the study. These two hospital clusters represent typical hospitals in Hong Kong with male and female closed wards where coercive methods are regularly used to manage patient aggressive events. In total, 14 wards were included in the study. The wards were either acute admission wards, psychiatric intensive care units or rehabilitation wards for adult in-patients. All wards used physical restraint on a daily basis. The number of hospital beds varied from 40 to 65, while the number of qualified nurses varied from 20 to 25 in each ward. In both hospitals, two of the most typical diagnoses were schizophrenia and mood disorder (Table 1).

### 2.2. Procedures

Restraint procedures and treatment are guided by the Hospital Authority [28], which provides general guidance for every psychiatric service in Hong Kong. The use of physical restrictions in Hong Kong psychiatric hospitals are divided into two levels (Level I, Level II) according to the Hospital Authority [28]. Both levels, Level I and Level II, can be applied to patients with voluntary or involuntary status. Physical restriction on Level I is part of the comprehensive management plan to help protect the patient from imminent health hazards when alternative and less-restrictive options are considered insufficient. This applies to situations such as preventing falls, supporting posture or the management of behavioral disturbances in patients who are cognitively impaired, or confused in nature, or those exhibiting constant behavioral manifestations that are either safety-threating, disturbing, disruptive, or self-harming in propensity and are refractory to drug treatment. The use of Level I physical restraint should always be based on the decision and regular review of the physician/multi-disciplinary team. Its intended duration is four hours or less. Its content should be discussed with relatives at the initial prescription and after a weekly review by the physician. Careful instructions are provided for staff caring for patients during physical restraint and when handling its documentation [28].

The use of physical restrictions on Level II is part of the comprehensive ‘de-escalation’ management plan for those patients who suffer from serious agitation, combativeness, or aggressiveness and have not responded adequately to other standard treatment. The aim is to protect the patient and others from imminent health hazards. Its intended use is two hours or less. Explanations and discussions with relatives should occur after each restraint. As is the case for Level I, the care of a patient during physical restraint and documentation is carefully instructed for Level II restraint [28]. In this study, cases of physical restraint under Level II have been analysed.

The main responsibility of taking care of patients’ physical and psychological needs during physical restraint rests on nurses. Nurses monitor patients’ vital signs and document their condition at least every 30 min [28]. The role of a psychiatrist (the case doctor) is to prescribe and authorize the physical restraint, and to review the patient regularly in terms of the restraint duration and to review medication. Psychologists or other professionals (social workers, for example) do not play any role in the restraint procedure. Debriefing patients or staff members during or after a restraint period is not a standard process or requirement in current Hong Kong treatment practices.

### 2.3. Data Collection

We screened all documents and extracted all incidents where physical restraints were used for patients in the study wards during the data-collection period. Wards where physical restraint is not in use, such as general psychiatric hospital wards, were excluded.

The data collection was focused on a specific time period between 1 July and 31 December 2018. The information about the use of patient physical restraint was extracted from hospital medical records using the specific data extraction tool developed for this study. The content of the tool was based on the Hospital Authority Guideline used in Hong Kong psychiatric hospitals [28] and the study wards’ patient-monitoring forms. This ensured that the data collected in both hospitals were comparable despite differences in documentation styles in both hospitals.

The following information related to each physical restraint event was collected from the physical restriction forms filled by nurses as part of patient medical records: description of the hospital (identification code, ward type), description of the patient (age, gender, legal status [voluntary and involuntary patients]), diagnosis, ICD-10, WHO [29], description of the event (date, specific time, type), physical restraint method used (safety west, waist belt, limb holder, other, e.g., magnetic shoulder trap), reason for the use of physical restraint (self-harm, violence, absconding, other), alternative interventions used before restraint use (discussed with each patient, time-out, de-escalation, diversional activities, other, sedation), and the length of the physical restraint (hours).

The contact persons in each hospital was responsible for the data collection because researchers outside the Hospital Authority did not have access to the medical records or any nurses’ notes. For the study purposes, the information from hospital records was also extracted by contact nurses using a specific data-extraction form designed for this study. The filled data-collection forms were returned to the researchers who developed an electronic database for further data analysis.

### 2.4. Data Analysis

Descriptive statistics (numbers [*n*]; frequencies [f], Mean, Standard Deviations [SD]) were used to describe each physical restriction incident. The physical restraint rate was calculated (i.e., the number of patients being restraint divided by the total number of patients). Patient age was categorised into four groups (30 or below, 31–40, 41–50, and 51 or above). The length of each incident was first described in one-hour periods (−60, 61–120, 121–180, 181–240, over 240 min) and further re-categorised into two groups based on the restraint time: two hours (120 min) or less and more than two hours (121 min and more); the length of each specific restraint method was then calculated. Associations between patient characteristics (age, gender, legal status, and diagnosis) and dichotomised physical restriction time were examined using Chi-square test. For occurrence, we calculated incidents on the ward level per total number of patients. The data analysis was conducted with IBM SPSS Statistics 26.

### 2.5. Ethical Considerations

Ethical approval was granted from the Human Subjects Research Ethics Committee of the Hong Kong Polytechnic University (ref: HSEARS20170206007). The Research Ethics Committee of both hospital clusters of the Hospital Authority also approved the study based on their ethical evaluation (HKECREC-2017-038; KW/FR-18-044(121-04)). The hospital staff anonymised personal information for each restraint event (patient name, personal code used by the hospital) before data management. The data were saved using ID code. All collected data were analysed and not recognisable at an individual level.

## 3. Results

### 3.1. Characteristics

Altogether, 1798 coercive incidents of physical restraint were recorded at the study hospitals between 1 July and 31 December 2018. At the same time as the data collection period, a total of 4170 patients were admitted to the study wards (Table 1); the rate of patient physical restraint events was 0.43. The mean age of all physically restrained patients was 40.0 years (SD 15.7, *f* = 1790). The gender distribution for events was equal (50% vs. 50%). The most typical diagnosis was schizophrenia-spectrum disorder and other psychotic disorders (61%). Typically, restraint events were faced by those who were treated voluntarily (70%, *f* = 1251). Detailed characteristics of events are provided in Table 2.

### 3.2. The Time of Physical Restraint Events

The specific time for each physical restraint incident was analysed. The most critical time points for physical restraints was late in the afternoon between 17:00 h and 18:00 h and later in the evening (20:00 h and 21:00). (Figure 1). The restriction times in the morning and evening were compared using Chi square test but no statistically significant differences were found (10:00–11:00 a.m. vs. 17:00–1800 p.m., chi-square = 2.52 (df = 1), *p*-value = 0.11; 10:00–11:00 a.m. vs. 20:00–2100 p.m., chi-square = 0.94 (df = 1), *p*-value = 0.33).

### 3.3. Types of Physical Restraint

Overall, 1986 physical restraints with different restriction types were recorded for 1798 incidents; the total number of restraints used is higher than the physical restraint incidents as more than one restraint type could be used during each incident. Different physical restraint types included a safety vest (*f* = 4), waist belt (*f* = 281), limb holder (*f* = 1690), and magnetic shoulder traps (*f* = 11) (Table 3).

The length of each restraint type for separate incidents was analysed. About half of all the physical restraint incidents took 1–2 h (52.23%), and seldom lasted less than one hour (5.24%). Only two incidents took more than four hours (Table 3).

### 3.4. Reasons for Physical Restraint and Alternative Methods

Overall, a total of 2122 reasons or indications were given for physical restraint events. As more than one reason could be attributed to each incident, the total number of reasons was higher than the total number of incidents. The reasons for physical restraint included violent behaviour (f = 990, 47%), self-harm (f = 122, 6%) and absconding (f = 109, 5%). Other reasons were also reported (f = 901, 42%), such as poor self-control, agitated mood, or displayed temper. Over 4517 alternative methods were used before physical restraint events. The most frequently mentioned alternative methods were an explanation of the situation to patients via discussions (1734/4517, 38%), time-out (1379/4517, 31%), or medical sedation (873/4517, 19%). De-escalation (422/4517, 9%) and other less coercive methods were also identified (78/4517, 2%). In addition, a few alternative methods with diversional activities were used (30/4517, 1%) including music, arts and crafts.

The analysis of the differences between the length (less than two hours vs. two hours or more) and any physical restriction type used generally showed statistically significant differences. The length of physical restriction incidents was longer (two hours or more) if a restrained person was younger, male, not treated on a voluntary basis, and if they were diagnosed with a neurodevelopmental disorder (Table 4).

## 4. Discussion

We compiled a register for the analysis of physical restraint incidents in two Asian psychiatric hospitals. Previous studies have shown variations in the types, frequency, and duration of restraint and seclusion across different countries and differences in the perception of restraint and seclusion between nurses and patients [30]. During the six-months data collection period, we registered 1798 physical restraint incidents. In our study, young age, male gender, being treated on a non-voluntary basis, and the diagnosis of a neurodevelopmental disorder were associated with a longer physical restraint period. This finding supports previous research findings in Japan [31] and Norway [32], where males have longer restraint periods compared to females. Our findings related to age are also consistent with the literature. In Norway, patients who are frequently restrained during their treatment period seem to be younger [33], and in China, restrained patients were younger compared to non-restrained patients [19]. Furthermore, some previous studies show that patients with a diagnosis of dementia (F00) and depression (F32) [34] or those persons diagnosed with schizophrenia (F20–F29) seem to be at most risk for prolonged restraint use [35]; which is at odds with our findings. On the contrary, we found similar results to authors in Israel who reported that patients who were diagnosed with schizophrenia tended to be restricted for a shorter period of time compared to patients with other diagnoses [36]. One reason for our results may be that treatment for patients with schizophrenia is well structured with an established medication regimen and is clearly defined in care plans because this patient group is often well-known to service providers. Further, Nieuwenhuis et al. [35] found that people with intellectual disabilities are at increased risk of being subjected to coercive measures during their inpatient care in the Netherlands. Although we did not make similar observations in relation to patients with significant limitations in intellectual functioning, we did find that patients with neurodevelopmental disorders seemed more likely to be restrained for longer periods. However, more studies are required to confirm whether this is really the case based on a larger sample and by utilising data obtained for a longer period of time.

We found some peaks in the use of physical restraint. The most critical time points for physical restraints was late in the afternoon and again later in the evening, which may be associated with ward routines, activities and staffing levels. At those specific times, patients might receive less attention from nurses which may increase uncertainty and disturbed behaviour. For example, new patients arrive on the wards at 5 p.m. and they may need extra attention due to their aggressive behaviour. Other peak times may be caused by the fact that patients are permitted to go for a shower or have dinner during specific times only, which may cause additional disruption, or staff members on the wards changing shifts or taking lunch breaks. The same finding has been reported in previous studies; patient aggression events increase when nurses rotate their work shift or at other times when wards are understaffed [24,37,38]. Kuosmanen et al. [38] further reported that although safety incidents in forensic psychiatric wards occurred at all times of the day, incidents involving violence against another patient peaked in the afternoon (14:01–16:00 h, 19%) and evening (18:01–20:00 h, 17.3%). This finding is also supported by the review of Weltens et al. [24] who found that patient aggressive events increased during ward rounds when nurses and doctors were occupied. Our study may therefore confirm that an awareness of the trend of patients being physically restricted might allow nurses to reorganise their schedules to increase their visibility on the wards, calm the atmosphere of the ward, and thereby prevent events which lead to the physical restriction of patients.

In comparison to studies outside Asia, we found that the duration of physical restraint incidents was rather short, as over half of the incidents lasted less than two hours. For example, in the international comparison study by Steinert el al. [10], the duration of mechanical restraints was, on average, 9.8 h in Germany, 11.1 h in Finland and 48 h in Japan. In contrast to some other studies, our results could be seen as positive if the reduced length of the restrictions result from staff’s constant assessment of the need for physical restraints, therefore suggesting that patients are not physically restrained unnecessarily for long periods of time. In a previous study by McKenna et al. [39], some patients in psychiatric wards were subjected to prolonged seclusion (>8 h, 55/206, 27%) and/or prolonged mechanical restraint (>1 h, 31/131, 24%) because staff sought to avoid ‘risk of harm to others’. Typically, most physical restraints in the current study were prompted by patient aggression, which supports the results of previous studies conducted in both Asian and Western settings [17,40]. Indeed, the priority in psychiatric care is to keep patients and others safe and to protect them from violence and associated types of safety incidents [41].

### 4.1. What the Study Adds to the Existing Evidence

In contrast to previous studies, our findings reveal that physical restraints were frequently used in incidents involving patients that chose to remain on the wards: 70% out of 1798 events of physical restraint occurred with patients who were treated on the ward on a voluntary basis. On the other hand, in general, our data showed that the duration of physical restraint was short, and it was longer for those patients who were treated against their will. This finding seems to be logical and supports the results of Pérez-Revuelta et al. [42] who showed that an involuntary status predicted the use of restraints. Still, from the human rights perspective, it is somewhat dubious that physical restraints are used so often on patients that are in voluntary care. The use of restraint should only be used in extreme situations where all other treatment options have failed [14]. In order to ensure that restraints are used as a last resort, there is an increasing need to report and understand patient safety incidents in psychiatric hospitals [43]. As already stated by Sashidharan et al. [44], it might also be time for a re-evaluation of current mental health legislation and practices to reduce coercion in psychiatric hospitals.

Our study builds on our previous data showing that comprehensive restrictive intervention-reduction programmes may be less used in Hong Kong [45]. Coercive practices, especially physical restraints, may be routinely used for patients, whether they have an in voluntary or non-voluntary status. Indeed, the rate of patient physical restraint events was 0.43 in the current study. In previous studies, the frequency of using physical restraint varied, from 4.5 to 9.4% for admitted patients in Europe [8] and has been higher in Mainland China (51.3% of all patients) [19], while the reform of mental health law in China seems to have led to a decreasing trend in the use of restraints in psychiatric hospitals (from 30.7% to 22.4%) [18]. In Germany, Mann et al. [46] recently reported that 8.0% of patients treated in four psychiatric hospitals were subjected to seclusion and/or mechanical restraints, and Välimäki et al. [47] found, in their national-wide register study in Finland, that 3.8% of patients had been mechanically restrained during their hospital stay.

### 4.2. Strengths and Limitations

Our study focuses on a topic which has often been ignored in Asian countries. Typically, studies related to patient physical restrictions in Hong Kong psychiatric hospitals have been related to staff’s or patient’s perceptions or attitudes toward the use of physical restraint [48,49] or factors associated with patients’ psychosocial and clinical variables [20]. In our study, for the first time, we described contextual factors and the events that occurred during physical restraints. Although the sample was limited and not based on patient electronic records, it still offered unique information about physical-restriction events based on authentical physical-restriction forms used in psychiatric hospitals.

Our study also had limitations, which need to be considered carefully. First, our data covers all cases of patient physical restrictions between 1 July and 31 December 2018. Our work could be complemented by an analysis of patient-level data to ascertain whether fewer disturbed patients have been physically restricted more than once due to their poor mental condition. Second, although the hospitals are part of Hospital Authority and follow their guidance, there were differences in documentation, ward structures, concepts used and also variations in how restrictions are monitored and reported in both hospitals. This raises a question as to whether information regarding patient restrictions are identified, collected and analysed in similar ways in different hospitals and whether the data are truly comparable between hospitals. Third, physical restrictions may vary depending on a variety of factors (e.g., patient mental status, day of hospitalization, the length of stay, medication etc.) and more information about patients would offer more light to this complex issue. Such information would have revealed any potential relationships between length of stay and restraint events. Fourth, we did not collect information about all admitted patients on the study ward during the data collection (i.e., patients who were not physically restricted). By comparing the specific characteristics of restricted and non-restricted patients we could have built a better picture of how such patients differ for all involuntary and voluntary admissions. Finally, we are not aware of how different research contexts, environments or cultures were associated with the results. In the future, an electronic recording system using the same information collected in all seven hospitals under the Hospital Authority could offer more reliable and usable data for quality assurance purposes and practice-development initiatives.

At the same time, it is important to interpret our results thoughtfully and cautiously as it remains unclear why during these extreme forms of patient restriction were used so frequently in psychiatric hospitals. It is important to keep in mind that alternative methods were documented in the study data, such as an explanation of the situation to the patients, time-out, and medical sedation, all of which indicate that alternative treatment methods are available and used in challenging situations. In addition, as guidelines and physical restraint guidelines may be context-bound, their content cannot be automatically interpreted or implemented in other sociocultural contexts without understanding similarities and differences between geographical and cultural areas.

However, as negative psychological complications of physical restraint and the compound effect of enforced medication can lead to powerful experiences which evoke shame, humiliation, and fear [50], all efforts should be made to avoid the unnecessary use of physical restraint. To prevent any aggressive events leading to the use of physical restraint, especially for patients with schizophrenia, different treatment options could be tested, including long-acting injectable antipsychotics [51,52].

### 4.3. Implications for Practice

Our analysis of incident reports has provided a general portrayal of the nature of physical restraint incidents in two psychiatric hospitals in Hong Kong. The analysis has improved our preliminary understanding of the circumstances in which incidents of physical restraint occur and has helped to identify to whom they occur, when, what types occur, for what reasons, and what efforts were made to prevent such incidents. These findings highlight that clinical practices in psychiatric wards should be analysed more deeply to identify any environmental and organisational factors in order to decrease any tension on the ward and thereby decrease the need for physical restrictions. We should further increase our understanding of why events of physical restraint mostly involve patients who are treated voluntarily in psychiatric hospitals in future research.

Regarding nurses’ schedules and actions on the ward, it is important to understand the rationale of nurse actions and timing—i.e., what is happening and when. Nurses could reorganise their tasks in order to prepare for times that are associated with more violent behaviours, such as new patients arriving on the wards. Nurses could also stagger their lunch breaks so that wards are not understaffed and increase the visibility of nurses during critical moments on the wards. In addition, the rationale to maintain restricted rules on the ward could be better developed with patients, to create a common understanding and achieve treatment goals on the wards. In addition, a systematic debriefing process with patients and staff members could be established as this seems to be missing in current treatment practices.

It is a well-known fact that nurses in Hong Kong are engaged in monitoring patients’ well-being using a high number of assessment methods. The procedures and treatment are guided by the Hospital Authority [28], which provides general guidance for every psychiatric service in Hong Kong. At the same time, differences in data-monitoring forms, concepts used, specific guidance in each hospital, and practical arrangements have raised a question as to how similarly or differently patients are really treated in different hospitals [45]. It might be time to put together specific hospital guidelines, monitoring forms, and assessment methods to understand what constitutes treatment in these hospitals. This study is one of the sub-studies in a series of studies aiming to understand treatment practices in Hong Kong psychiatric hospitals. It might also be time to analyse more deeply the current clinical practices in psychiatric hospitals across Hong Kong and look towards developing more humane and restriction-free service environments by focusing on staff training, risk assessments, psychotherapy, debriefings, and advanced directives [53].

## 5. Conclusions

The current study shows that physical restraints are frequently and routinely used in Hong Kong psychiatric inpatient care. This study also revealed ethically problematic areas in the use of restraints, namely the use of physical restraints on people who are admitted into the psychiatric wards on a voluntary basis and on people with neurodevelopmental disorders. Monitoring data regionally and locally could reveal potentially problematic areas, which might further help to highlight areas where actions are most needed.

## Figures and Tables

**Figure 1 ijerph-19-06032-f001:**
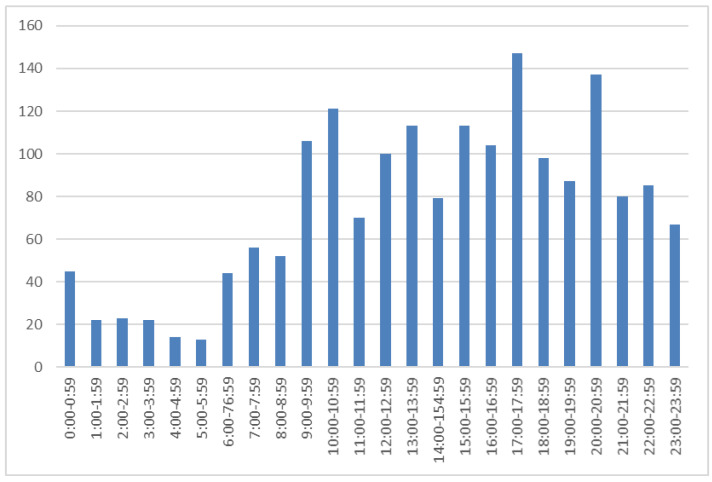
The specific time (h) of each restraint that occurred on the wards (*f* = 1798).

**Table 1 ijerph-19-06032-t001:** Characteristics of the study wards (based on the hospital statistics on year 2018).

	Hospital A	Hospital B
Number of study wards	6	8
Specialty of the wards	4 acute admission wards 2 rehabilitation wards	Intensive care wards for acute care
Gender type of the wards	3 female wards 3 male wards	4 female wards 4 male wards
A range of a number of beds in each study ward	40–48 beds	50–65 beds
Number of staff working on the study wards	Around 20 nurses per ward	About 25 nurses per ward
Typical age distribution of patients	18–65 years	16–64 years
Two most typical diagnoses	Schizophrenia Mood disorder	Schizophrenia Mood disorder
Number of patients treated on the study wards during the data collection	913	3257

**Table 2 ijerph-19-06032-t002:** Characteristics of the physical restraint events analysed.

	*f*	%	Mean (SD)
**Physical restraints (*f* = 1798)**			
Hospital A	616	34	
Hospital B	1182	66	
**Patient age in each incident (*f* = 1790)**			39.96 (15.73)
Range 13–95 years			
**Patient gender in each incident (*f* = 1798)**			
Male	899	50	
Female	899	50	
**Patient legal status in each event (*f* = 1593)**			
Voluntary	1251	70	
Non-voluntary	342	30	
*** Diagnosis, ICD-10 ^1^ (*f* = 1106)**			
Neurodevelopmental disorder	199	18	
Schizophrenia spectrum and other psychotic disorder	672	61	
Affective/mood disorder	151	13	
Substance-related addictive disorder	32	3	
Other	52	5	

* Primary diagnosis; ^1^ World Health Organization 1993; f = frequency; % = percentage; SD = standard deviation.

**Table 3 ijerph-19-06032-t003:** The length of each physical restriction and restriction method used (N = 1798).

	Safety West		Waist Belt		Limb Holder		Magnetic Traps		Total	
	*f*	%	*f*	%	*f*	%	*f*	%	*f*	%
Minutes										
−60	0	0	8	8	95	91	1	1	104	5.24
61–120	4	0.4	163	15.1	906	84.1	4	0.4	1077	52.23
121–180	0	0	24	12	175	88	0	0	199	10.02
181–240	0	0	86	14	512	85	6	1	604	30.41
241-	0	0	0	0	2	100	0	0	2	0.1
Totally	4		281		1690		11		1986	

*f* = *frequency*; % = percentage.

**Table 4 ijerph-19-06032-t004:** The differences between restrictive incidents and different restrictions times (less than two hours vs. two hours and more).

	Total		<120 *		120 *≤		
	*f*	*f*	%	*f*	%	Chi Square (df)	*p*
**Variables**							
**Age**						9.11 (3)	0.028
30 or below	642	373	58.1%	269	41.9%		
31–40	295	165	55.9%	130	44.1%		
41–50	349	215	61.6%	134	38.4%		
51 or above	504	329	65.3%	175	34.7%		
**Gender**						111.47 (1)	<0.001
Male	899	433	48.2%	456	50.7%		
Female	899	652	72.5%	247	27.5%		
**Legal status**						10.61 (1)	0.001
Voluntary	1251	786	62.8%	465	37.2%		
Other	547	299	54.7%	248	45.3%		
**Diagnosis**						30.04 (4)	<0.001
Neurodevelopmental disorder	199	79	39.7%	120	60.3%		
Schizophrenia spectrum and other	672	404	60.1%	268	39.9%		
Affective/mood disorder	151	90	59.6%	61	40.4%		
Substance-related addictive disorder	32	16	50.0%	16	50.0%		
Other	52	35	67.3%	17	32.7%		
* Minutes							

* Minutes; *f* = *frequency*, % = percentage; *p* = *p*-value.

## Data Availability

Data are available upon request due to privacy restrictions.

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
