# Peer review of "Physical Restraint Events in Psychiatric Hospitals in Hong Kong: A Cohort Register Study"

_ijerph, 2022, doi:10.3390/ijerph19106032_

Round 1
Reviewer 1 Report
The paper is well written and the methods are sound.
Introduction:
The introduction gives an appropriate overview over the topic and the aim of the study.
In line 43 it remains unclear of which sample of people percentages (4.5-9.4%) are meant. (patients, population..) please clarify
Methods:
Line 192: What kinds of physical restraint methods were categorized for the analysis? Please specify.
Results:
Table 2: Were patients characterized by main diagnosis or can patients occur as well in the group of schizophrenia spectrum disorder as in the group of substance related disorder when having both?
Please add a caption to the table.
Table 4:
It would be interesting to know when during the hospital stay the restrictive incident occurred (which day of hospitalization, and information about length of stay if available)
Discussion:
As reduction of physical restraint is a common goal it might be interesting to discuss influence of medication on reduction of restraint especially in patients with schizophrenia spectrum disorder which is the biggest subgroup in this study.
Early treatment of psychosis is a relevant factor in reducing and avoiding physical restraint. It would also be interesting to discuss if patients treated with Long-Acting Injectables show lower rates of restrictive incidents as there is a concensus that LAI should be used more often. Rates of use of Long Acting Injectables in acute psychiatric inpatients in Europe have recently been published. (doi: 10.3390/jpm12030441).
I would like to add the following citations for the point of inpatient LAI use:
doi: 10.1155/2019/8629030
doi: 10.18553/jmcp.2015.21.9.754
Reviewer 2 Report
This is an interesting study that addresses an important topic, that is the application of restraint in inpatient mental health treatment. The paper is well-written and informative for the national and international readership.
I have some suggestions for the authors to improve the paper.
The authors report that 70% of restraint incidents involved voluntarily admitted patients. They do not provide the percentage of the total number of voluntarily admitted patients among the total of 4170 admissions. I think it would be more informative to compare the percentages of involuntary vs voluntary admitted patients that had been subjected to restraint in relation to total involuntary and voluntary admissions. It has been repeatedly shown that involuntary status at admission is related to restraint and seclusion. If relative data are not available, the authors should mention this limitation.
See for example:
Bilanakis, N., Kalampokis, G., Christou, K., & Peritogiannis, V. (2010). Use of coercive physical measures in a psychiatric ward of a general hospital in Greece. International Journal of Social Psychiatry, 56(4), 402–411.
Cole, C., Vandamme, A., Bermpohl, F., Czernin, K., Wullschleger, A., & Mahler, L. (2020). Correlates of seclusion and restraint of patients admitted to psychiatric inpatient treatment via a German emergency room. Journal of Psychiatric Research, 130, 201–206.
Odawara, T., Narita, H., Yamada, Y., Fujita, J., Yamada, T., & Hirayasu, Y. (2005). Use of restraint in a general hospital psychiatric unit in Japan. Psychiatry and Clinical Neurosciences, 59(5), 605–609.
There are some grammar and syntactical errors that could be easily corrected e.g lines 42-43, line 91 (large numbers of studies, you can change to: many studies), line 98 (check the word used), line 104 (health care industry?).
Please make sure that table 2 appears in one page, it is divided in this draft
Reviewer 3 Report
ID: ijerph-1682056
Title: Physical Restraint Incidences in Psychiatric Hospitals in Hong Kong: A Retrospective Longitudinal Cohort Register Study.
Thank you for the opportunity to review this paper. Despite it is an interesting and well-down manuscript, I still have some recommendations for you to make it more suitable for publication.
Comment:
Major revision.
Detailed information:
Abstract:
Page 1, line 22: since you said “incidents” yourself, I do not think “prevalence” is a proper word. Other than that, I think your abstract could be written more logically.
Introduction
Page 3, line104-106: What is innovative about this research? What are the practical implications of understanding these conditions?
Materials and Methods
Design and Setting
Page 3, line 117-120: Your data only covered 6 months, I do think such a short time could considered as a longitudinal study.
Page 3, line 130, Table 1: I have following questions for Table 1.
First, are schizophrenia and mood disorder the two most typical diagnoses in the fourteen wards or in the two hospitals? Second, if it belongs to the former case, can you tell me why you chose these two rather than other diagnoses in psychiatry as the subject of your study? What’s more, the specialty of the wards is inconsistent. In consequence, are factors such as the time occurred and the duration of the physical restrictions affected by this? Can they be put together to draw conclusions? Finally, “Two most typical diagnosis” should be changed into “Two most typical diagnoses”.
Page 3, line 168-169: How to be retrospective since all the records you listed below are not "retrospective"?
Page 3, line 177-182: Was the data extracted anonymously. How could the data be secured since all your researchers outside the hospital?
Procedures
Page 3: Two levels of physical restrictions are introduced in detail and I do get the differences between two, however, I don’t think you only describe the levels in this part is correct. “Procedures” need present more details of the entire research process.
Data Analysis
Page 5: Which software was used in the data analysis?
Results
Characteristics of Events Regarding Physical Restraints
Page 5, Table 2: What does “f” stand for? What does “%” stand for? What does “SD” stand for? Every full name of the abbreviation should be shown in “Notes” below the table. In addition, you’d better adjust column names.
The Time of Physical Restraint Incidents Occurred
Page 6, line 217-219: I found that the frequency of occurrence from 10:00 to 11:00 is also high, so is the difference statistically significant compared to the two time periods you mentioned?
Page 6, line 220-221, Figure 1: The title upon the bar chart is not required.
Types of Physical Restraint Incidents
Page 7, line 232¸Table 3: The "%" is missing from the "Magnetic traps" column. Furthermore, it seems that the tables in this article were not standardized and formatted.
Page 8, line 252-253, Table 4: since your subgroups were “<120” and “>121”, how is your “120” should be categorized? Where is it?
Discussion
Page 8: The main findings should be interpreted in the same order as the "Results" section.
Page 9, line 301-303: Your findings is at odds with similar studies before, please explain the possible reasons that could lead to this result.
Page 9, line 308-311; Page 10, line 366: Can you draw the conclusion that patients with neurodevelopmental disorders seemed more likely to be restrained for longer periods only based on this study? There are only two typical diagnoses, what about other diagnoses in psychiatry?
Page 10, line 341-344: “Our work could be complemented by analysis of patient-level data to ascertain whether fewer disturbed patients have been physically restricted more than once due to their poor mental condition.” Did you try to solve the limitation?
Conclusion
Page 11, line 396-397: Does this study really have any innovation or meaning? I doubt it since you do not even conclude your “strengths”.
Taking all the comments above into consideration, this research is interesting and meaningful. However, from my perspective, the data should be further analyzed and you may get more useful information from it. Moreover, some of your subtitles are way too long, shorten them please. In addition, the quality of written English could be improved by consulting a native English speaker.
My bests,
Your reviewer
Reviewer 4 Report
The aim of the study was to describe physical restraint incidents in psychiatric hospitals reported by hospital staff in the reporting system during a hospital stay.
The paper seems to be a fine contribution to this area of involuntary intervention, as monitoring is vital in measuring and improving the services.
The paper should be improved on certain issues.
Firstly, the abstract is written very informally, making it hard to grasp quickly, particularly the conclusion.
The formal decisions and local regulations for restraints are very important due to several reasons but are here rather vague described.
The authors write about physicians without mentioning more precise the role of the psychiatrists (and/or psychologists), which presumably are involved.
The relationship between physical restraint and medical intervention is often essential, either as a combined intervention or the one substituting the other. This aspect is mentioned but should be elaborated.
There is no mention of having a dialogue about the patients’ and the staff’s experience with the restraint interventions after the interventions, which would improve the ethical aspects of the physical restrains. If this item was not a part of the form, it should be in the next version and mentioned in the discussion.
This study was part of a clinical R&D project. The authors should mention practical translation ideas about implementing monitoring,integrated, as part of the routine clinical work.
I did not find many language errors but some minor (as: “About half of 228 the all physical restraint…”) and found the following sentence hard to understand: “We found a peak in the use of physical restraint in any time.”
Round 2
Reviewer 3 Report
The concern has been addressed. Good job!